# Application of Bispectral Index System (BIS) Monitor to Ambulatory Pediatric Dental Patients under Intravenous Deep Sedation

**DOI:** 10.3390/diagnostics13101789

**Published:** 2023-05-18

**Authors:** Shih-Chia Chen, Chun-Yu Chen, Shih-Jyun Shen, Yung-Fong Tsai, Yu-Chen Ko, Li-Chuan Chuang, Jr-Rung Lin, Hsin-I Tsai

**Affiliations:** 1Department of Anesthesiology, Chang Gung Memorial Hospital, Linkou Branch, Taoyuan 333, Taiwan; 2College of Medicine, Chang Gung University, Taoyuan 333, Taiwan; 3Graduate Institute of Clinical Medical Sciences, Chang Gung University, Taoyuan 333, Taiwan; 4Department of Anesthesiology, Chang Gung Memorial Hospital, Chiayi Branch, Chiayi 613, Taiwan; 5Department of Pediatric Dentistry, Chang Gung Memorial Hospital, Linkou Branch, Taoyuan 333, Taiwan; 6Department of Dentistry, School of Dentistry, National Yang-Ming University, Taipei 112, Taiwan; 7Graduate Institute of Craniofacial and Dental Science, College of Medicine, Chang Gung University, Taoyuan 333, Taiwan; 8Clinical Informatics and Medical Statistics Research Center (CIMS) and Graduate Institute of Clinical Medical Sciences, Department of Biomedical Sciences, Gung Gung University, Taoyuan 333, Taiwan

**Keywords:** pediatric dentistry, sedation, anesthesia recovery period, consciousness monitors, anesthesia, intravenous

## Abstract

**Purpose** Intravenous sedation has been well accepted to allow dental restoration in uncooperative children while avoiding aspiration and laryngospasm; however, intravenous anesthetics such as propofol may lead to undesired effects such as respiratory depression and delayed recovery. The use of the bispectral index system (BIS), a monitoring system reflective of the hypnotic state, is con-troversial in the reduction in the risk of respiratory adverse events (RAEs), recovery time, the in-travenous drug dosage, and post-procedural events. The aim of the study is to evaluate whether BIS is advantageous in pediatric dental procedures. **Methods** A total of 206 cases, aged 2–8 years, receiving dental procedures under deep sedation with propofol using target-controlled infusion (TCI) technique were enrolled in the study. BIS level was not monitored in 93 children whereas it was for 113 children, among which BIS values were maintained between 50–65. Physiological variables and adverse events were recorded. Statistical analysis was conducted using Chi-square, Mann Whitney U, Independent Samples t and Wilcoxon signed tests, with a *p* value of <0.05 considered to be statistically significant. **Results** Although no statistical significance in the post-discharge events and total amount of propofol used was observed, a clear significance was identified in periprocedural adverse events (hypoxia, apnea, and recurrent cough, all *p* value < 0.05) and discharge time (63.4 ± 23.2 vs. 74.5 ± 24.0 min, p value < 0.001) between these two groups. **Conclusions** The application of BIS in combination with TCI may be beneficial for young children undergoing deep sedation for dental procedures.

## 1. Introduction

Childhood dental caries is common and multifactorial, with poor oral hygiene, bad eating habits, and changes in oral bacteria as major contributors [1]. Risk assessment and prevention protocols based on dietary habits are important, but some may still require intensive treatment such as extractions and restorations in early childhood. Cooperation during dental procedures is critical, especially for children who may have had unpleasant experiences before. The potential psychological trauma could contribute to dental anxiety in adulthood [2]. For these patients, sedation can be an efficient method for teeth restoration while minimizing pain and anxiety. In recent years, there has been a significant rise in dental procedures being performed on pediatric patients outside conventional operating rooms. This shift, along with increased public awareness of pain management and anxiety reduction, has led to a demand for sedation in outpatient dental settings [3].

The level of sedation is a continuum, ranging from minimal sedation and moderate sedation, in which patients can respond to verbal commands and maintain their own respiratory and cardiovascular function, to deep sedation and general anesthesia, in which patients are not arousable with possible impaired ventilatory function [4]. To achieve successful outcomes, it is essential to consider a child’s age and emotional development when providing adequate sedation. For example, minimal sedation has become widely accepted as a safe and effective method for fearful patients who require complex dental procedures [5]. Anxiolytics are often given to help patients relax and reduce anxiety during the procedures while remaining responsive to the practitioner. However the techniques commonly employed in light sedation, particularly oral sedatives, can be challenging to predict the sedation level and often result in a certain degree of failure rate [6]. Procedural sedation, or Monitored Anesthesia Care (MAC), also aims to help anxious patients get through the procedures while avoiding obnoxious memories [7]. Nevertheless, the depth of sedation can vary as the stimulation varies during different stages of the procedures. Particularly, preschoolers often struggle to tolerate such procedures under light sedation as the ability to control his or her own behavior to cooperate for a procedure depends on the age and cognitive development. Therefore, deep sedation or general anesthesia may be preferable for children of very young age [8].

Deep sedation involves drug-induced consciousness depression, where patients are not easily aroused but able to respond purposefully to repeated verbal or painful stimuli. Sometimes, spontaneous ventilation may be impaired and when this occurs, and assistance to maintain a patent airway is necessary. Concerns in regards to ventilation include decreased breathing rate or depth which may cause a reduction in blood oxygenation and lead to complications such as respiratory failure or cardiac arrest [9]. Cardiovascular function, on the other hand, is typically preserved, but protective airway reflexes may be lost, potentially transitioning to general anesthesia. Propofol has been the choice of sedative in various clinical scenarios due to its quick onset and brief recovery period. Its anti-emetic properties, absence of emergence agitation, hemodynamic stability, and hypnotic effectiveness further contribute to its appeal as a sedative option for children [10,11,12]. That said, propofol may cause respiratory depression, upper airway obstruction, hypoventilation and a subsequent decline in oxygen saturation. It is imperative for the practitioner to have the ability to identify and manage these known effects to ensure optimal safety [13].

Given the potential risks involved, deep sedation is reserved for invasive or complex procedures that cannot be performed under lighter sedation or local anesthesia. The risks vary depending on the patient’s medical history, medications used, and the procedure type. Children under five and those with complex medical backgrounds face the highest risk [14]. Close monitoring of vital signs and careful sedation management are crucial to minimize these risks. Guidelines from the American Academy of Pediatrics and the American Academy of Pediatric Dentistry (AAPD) recommend monitoring the depth of sedation during dental procedures [15]. Several subjective assessment tools have been developed, such as the Richmond Agitation Sedation Score (RASS) [16], the Ramsay Sedation Scale [17], the Observer’s Assessment of Alertness Sedation Scale (OAA/S) [18] and Pediatric Sedation State Scale (PSSS) [19]. However, these scores were often subjective, excitatory, and disruptive, and can be difficult to carry out for uncooperative children under sedation. Significant efforts have been dedicated to developing a consistent method for maintaining anesthetic levels to enhance patient safety during sedation and anesthesia procedures.

Target-controlled infusion (TCI) is a technique that uses computerized algorithms to manage the infusion rate to achieve the required plasma concentration or brain concentration, referred to as the “target concentration” [20]. As an open-loop system, TCI offers reliable control over anesthesia depth, enabling healthcare professionals to adjust infusion rates based on the patient’s response [21]. Several studies such as that by Casati et al. in 1999 have attempted to determine the average target concentration (Cpt) for different levels of sedation [22]. The device calculates anesthetic dosage and concentration precisely based on pharmacokinetics, pharmacodynamics, height, weight, age, and gender. It also estimates recovery concentration based on the loss of consciousness concentration. TCI minimizes over-sedation or under-sedation risks, promotes quick recovery, reduces adverse effects, and enhances anesthesia safety [23]. However, it has drawbacks, such as potential inaccuracies due to individual variations, technical malfunction risks, and reliance on continuous monitoring. Prior research has indicated that older patients and those with a lower BMI are at a higher risk for hypotension. Additionally, extended procedure durations have been identified as a risk factor for both desaturation and hypotension [24].

Bispectral index (BIS) monitoring is a system developed to assess patients’ consciousness levels during sedation and anesthesia based on their electroencephalogram (EEG). BIS allows precise sedation level control without intentionally stimulating sedated patients [25]. It generates a score from 0 to 100, with higher scores indicating increased consciousness. Since 1997, BIS has been in clinical practice and a wealth of experimental research has accumulated on its use. Studies have demonstrated a strong correlation between BIS and subjective sedation scores in children [26,27,28,29]. The use of BIS monitoring has been shown to improve patient outcomes and reduce complications associated with sedation and anesthesia, such as respiratory depression or cardiovascular complications [30,31,32]. BIS monitoring may also help reduce drug dosage, which may further result in faster recovery after procedures [33,34].

Our primary objective is to determine whether BIS can reduce complication rates and minimize post-procedural adverse events in the pediatric dental population. The secondary objective is to assess whether BIS can decrease the total amount of propofol dosage administered and expedite recovery time.

## 2. Methods

The study protocol was approved by the Chang Gung Memorial Hospital Institutional Review Board (CGMH IRB: 202300491B0). Our retrospective study was conducted on pediatric patients aged 2–8 years who underwent dental procedures under moderate-to-deep sedation in our outpatient dental clinics between October 2017 and April 2019. The procedures included dental exams, cleanings, fluoride treatments, sealants, fillings, crowns, extractions, root canal treatments, and periodontal treatments. In this study, children presenting with known airway abnormalities, severe respiratory or systemic diseases, or allergies to anesthetics were excluded from participation. The research ultimately included a total of 206 children. Among these participants, 113 were monitored using the bispectral index (BIS) during the procedure, forming the BIS group, while the remaining 93 children were not monitored with the BIS, constituting the non-BIS group (Figure 1). As our study is a retrospective chart review, the use of the BIS monitor was not randomized and was left to the discretion of the anesthesiologist in charge.

### 2.1. Deep Sedation Protocol

As a standard practice, a thorough assessment was conducted for each patient before the dental procedure. This included evaluating their medical, surgical, and anesthetic history, as well as determining their pre-procedural fasting status. We have reviewed the documented demographic information, physical status, peri-procedural airway adverse events, administered drug doses, procedure time, time to awakening and to discharge, and post-procedural events within the first 24 h.

For the purpose of reducing anxiety, patients received oral premedications such as midazolam, ketamine, or chloral hydrate before undergoing venipuncture. Upon establishing the intravenous route, propofol infusion was initiated in all children using a TCI (Alaris^®^ PK Syringe Pump Propofol-Paedfusor Model) system. The goal for achieving deep sedation was to reach a Richmond Agitation-Sedation Scale (RASS scale) of −4 in both groups, which is a level of deep sedation characterized by no response to stimulation and minimal-to-no reflex activity. In the BIS group, propofol was titrated to maintain the ideal BIS range of 50 to 65, while in the non-BIS group, propofol was titrated as per the duty anesthesiologist’s discretion. The dentist provided local anesthesia using a 1% lidocaine solution combined with epinephrine prior to performing any potentially painful procedures in all patients. If patients displayed signs of stimulation reactions, such as body movement, increased heart rate, or elevated blood pressure, both groups received either local anesthetics or adjustments to the propofol dosage. In cases where these measures proved insufficient, intravenous anesthetics such as fentanyl, alfentanil, or ketorolac were administered to maintain a stable anesthetic state.

Once achieving a stable sedation depth, the patient was repositioned in the head tilt and chin lift posture, and a nasopharyngeal airway was routinely placed to ensure airway patency. Supplemental oxygen was administered through a nasal cannula (2–4 L/min). Standard monitors, including a pulse oximetry, heart rate, non-invasive blood pressure, and nasal capnography (EtCO_2_), were used throughout the procedure. During the procedure, rubber dams and gauze packing were used to minimize the collection of sprayed water and secretion into the airway. The anesthesiologist would record any RAEs during the procedure, including apnea, cough, wheezing, and stridor. Any necessary interventions such as repositioning, suction, mask ventilation, or endotracheal tube placement were also documented. Hypoxia was defined as a decrease in oxygen saturation level to SpO_2_ < 90% at any time, while apnea was defined as a respiratory pause lasting longer than 10 s [35]. Bradycardia was defined as a heart rate less than 60 beats per minute and hypotension as systolic blood pressure (SBP) less than 70 plus the person’s age in years multiplied by 2 (SBP < 70 + age (year) × 2).

When regaining consciousness after the dental procedure, patients were transferred to the post-anesthesia care unit (PACU), where they received close monitoring every 15 min until the established discharge criteria of AAPD was fulfilled. Postoperative adverse events within 24 h were collected by calling parents 1–3 days after the dental procedure to discuss any issues experienced during recovery.

### 2.2. Statistical Analysis

The demographic data was reported as mean ± standard deviation. An independent investigator analyzed the statistical data using the Windows version of IBM SPSS 24 (SPSS Inc.). Continuous variables were analyzed using independent t-tests and presented as mean ± SD while categorical variables were analyzed using Chi-square tests and presented as numbers (%). A *p* value of <0.05 was considered statistically significant.

## 3. Result

The demographic and medical characteristics of the 206 children were presented in Table 1. The average age of the patients was 45.9 ± 13.4 months, and the sedation time ranged from 30 min to 290 min with an average time of 119.4 ± 41.5 min. No statistical significance was observed in the children’s age, weight, American Society of Anesthesiology classification (ASA) status and total anesthetic time. Unintentionally, a higher number of female patients were found in the BIS group compared to the non-BIS group.

### 3.1. Primary Outcome

The incidence of adverse events observed during the dental procedure is summarized in Table 2.

We observed 132 episodes of perioperative respiratory adverse events (RAEs), with hypoxia being the most common, occurring 47 times, followed by apnea (38), recurrent coughing (16), bronchospasm (3), and laryngospasm (2).

In comparison to the non-BIS group, the BIS group showed significantly lower incidences of hypoxia, apnea and coughing (*p* value of 0.001, 0.002 and 0.017 respectively). No statistical significance was observed in the incidence of other RAEs, assisted airway maneuvers or cardiovascular events including bradycardia and hypotension.

A patient in the BIS group required intubation due to unceasing cough with severe desaturation, while in the non-BIS group, one intubation was required due to laryngospasm. The patient in the BIS group who required mask ventilation had prolonged apnea with desaturation, while in the non-BIS group, five patients required mask ventilation due to laryngospasm (1), recurrent cough (2), and desaturation (1). None of these patients experienced bradycardia or hypotension, and all patients completed their dental procedure and recovered afterwards.

Table 3 shows that the most frequently reported adverse event following the dental procedure was toothache, occurring in 87 cases. Within 24 h of discharge, 28 patients reported fever, 21 reported sleepiness, 14 reported dizziness, and six reported nausea or vomiting. No statistical significance was observed between the BIS and non-BIS groups.

### 3.2. Secondary Outcome

Table 4 displays the results of the study’s propofol usage and patient recovery times during sedation. The average propofol dose administered was 341 ± 135 mg, with a mean infusion rate of 0.178 ± 0.032 mg/kg/min. The average time for patients to awaken and discharge was 31.3 ± 17.2 and 68.4 ± 23.6 min, respectively. Time to awakening was defined as the duration from discontinuing propofol infusion to the point at which the patient became responsive to verbal commands.

No statistical significance was found in the total anesthetic time, time to awakening, total propofol dosage, or average infusion rate per kilogram. However, it was observed that the non-BIS group required a longer recovery time before being discharged (63.4 ± 23.2 vs. 74.5 ± 24.0 min, *p* = 0.001 respectively).

## 4. Discussion

Our findings confirmed that propofol-based sedation could be used safely and effectively in young children who were unable to cooperate during dental procedures, consistent with the previous literature [8,36,37,38]. The use of propofol has become increasingly popular for moderate-to-deep sedation in office-based procedures over the past few decades. The literature has revealed that propofol has a narrow therapeutic window and that propofol-based sedation may lead to hypoxia or other airway-related complications, which may result in rapid conversion to general anesthesia [15,16,17]. With advanced technologies, automated dosing with propofol-based target-controlled infusion (TCI) has been shown to provide accurate and stable predicted plasma and effect-site concentrations [21]. Additionally, the development of EEG-based technology such as bispectral index (BIS) monitoring has provided a direct measure of anesthetics and sedating effects on the brain, indicating the depth of sedation level [25]. Studies have demonstrated a strong correlation between TCI and the BIS index [39]. A combination of BIS monitoring with TCI in adolescents and adults receiving dental care leads to a significant reduction in the propofol dose and recovery time [40,41,42]. Based on the available evidence, we hypothesized that the combination of TCI and BIS monitoring may offer potential benefits for improving the safety and efficacy of sedation in pediatric dental patients.

During deep sedation in pediatric dental patients, the incidence rate of intraoperative airway complications ranges from 6.8% to 30.1% [28,29,30]. To minimize the risk of airway complications during dental procedures, it is crucial to maintain an appropriate level of sedation. Real-time monitoring of sedation depth using BIS during propofol-based sedation may help to prevent oversedation, thereby minimizing airway complications and hemodynamic instability [43]. Our research demonstrated a significant reduction in RAEs in the BIS-monitored group compared to the non-BIS-monitored group, consistent with previous studies reporting decreased complications in adults using BIS during procedural sedation [30,44]. While our findings did not demonstrate a significant decrease in cardiovascular events such as hypotension or bradycardia, BIS monitoring may still have potential benefits for improving the safety of sedation in pediatric patients, similar to that of adults.

The overall Incidence of postoperative nausea and vomiting in our study (2.9%) was similar to the literature on deep sedation, much lower than that reported by general anesthesia (13.7%) [45,46,47]. Despite that, no significant differences in other postprocedural complications between the BIS-monitored group and the non-BIS-monitored group were observed. These findings suggest that postoperative symptoms following dental procedures are common, but the occurrence may not be influenced by BIS monitoring.

Whether the use of BIS may or may not help reduce anesthetics consumption is controversial. Although some studies have reported benefits of BIS monitoring in a reduction of propofol dosage during dental procedures [30,40,41], Dag et al. [48] reported no significant differences in the amount of propofol used, incidence of adverse events, or recovery time between BIS-monitored and non-BIS-monitored pediatric dental patients. In our study, we did not find a significant reduction in the propofol dosage or awakening time but we observed a significant reduction in the recovery time before discharge between the BIS-monitored group and the non-BIS-monitored group. We have also demonstrated the benefits of BIS utilization in reducing the risks of peri-procedural adverse effects. These findings are consistent with previous studies that suggested BIS monitoring may lead to a faster discharge time in pediatric dental patients receiving general anesthesia [49]. Although the present study has established the benefits of BIS monitoring for pediatric dental procedures, it is not without limitations. The retrospective nature of our study precluded an identical protocol for oral premedication, and the use of local anesthetics by the dentists and supplemental analgesics by anesthesiologists, which may influence the sedation level. These results highlight the need for further research to investigate the potential benefits and limitations of BIS monitoring in different patient populations and clinical contexts. Further larger randomized controlled trials are warranted to validate these results across different sedation levels, age groups, and anesthetic agents.

Dental restoration has benefited children in their daily life; however, precautions in reducing bacterial load before dental anesthesia may further add benefits for these children. The use of natural substances such as probiotics, paraprobiotics, postbiotics, and ozonated gels may restore balance in the oral microbiota and reduce bacterial load [50]. Using these natural substances in pre- and post-sedation procedures may help to reduce the risk of infections by promoting a healthy oral microbiota and inhibiting the growth of harmful bacteria as evidenced in the literature.

## 5. Conclusions

Childhood dental caries is common and multifactorial, with poor oral hygiene, bad eating habits, and changes in oral bacteria as major contributors. Intensive treatment such as extractions and restorations may sometimes be necessary to improve these children’s overall health. That said, early education on dental hygiene is always important for young children. When dental restoration is employed in uncooperative children, sedation is sometimes essential, and the issue of safety must be emphasized. Through comprehensive personnel training, the development of sedation standards and protocols, as well as the introduction of innovative sedatives and new techniques for monitoring sedation levels such as BIS and TCI, outpatient sedation has become more effective and safer. Our study demonstrated that the use of the BIS monitor in combination with TCI propofol in moderate-to-deep sedation may benefit young children during dental procedures. Of particular interest, a reduction in airway-related adverse events and recovery time before discharge is of paramount significance for this population. By monitoring the depth of sedation, dental procedures can be completed under adequate sedation while minimizing the risk of respiratory-related adverse events in an office-based environment. Although the use of better monitoring can aid in providing safer anesthesia care, it is important to note that the anesthesia provider should always be well-trained and prepared for situations when endotracheal intubation or cardiopulmonary resuscitation is required.

## Figures and Tables

**Figure 1 diagnostics-13-01789-f001:**
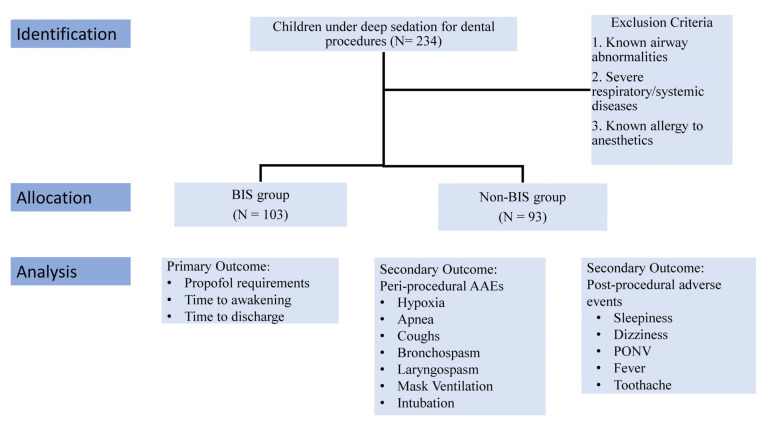
Flow chart of patient identification, allocation and analysis.

**Table 1 diagnostics-13-01789-t001:** Demographics.

	Total (N = 206)	BIS Group (N = 113)	Non-BIS (N = 93)	*p* Value
Age (month)	45.9 ± 13.4	46.3 ± 12.8	45.5 ± 14.1	0.676
Weight (kg)	16.34 ± 3.4	16.35 ± 3.5	16.33 ± 3.4	0.966
ASA classification (I/II)	151/55	77/36	74/19	0.082
Gender (Male/Female)	125/81	60/53	65/28	0.015 *
Anesthetic time (min)	119.5 ± 41.5	121.8 ± 45.7	116.5 ± 35.5	0.357

Data expressed as mean ± SD or number. BIS, bispectral index; ASA, American Society of Anesthesiologists; min, minute; mg, milligram; kg, kilogram. * denotes statistical significance with *p* value < 0.05.

**Table 2 diagnostics-13-01789-t002:** Events during the dental procedure.

	Total Number (%)(N = 206)	BIS Group (%)(N = 113)	Non-BIS (%)(N = 93)	*p* Value
Peri-procedural RAEs				
Hypoxia	47 (22.8)	16 (14.2)	31 (33.3)	0.001 *
Apnea	38 (18.4)	12 (10.6)	26 (28.0)	0.002 *
Recurrent Cough	16 (7.8)	4 (3.5)	12 (12.9)	0.017 *
Bronchospasm	3 (1.5)	3 (2.7)	0 (0)	0.253
Laryngospasm	2 (1.0)	0 (0)	2 (2.2)	0.203
Assisted airway maneuvers				
Endotracheal Intubation	2 (1.0)	1 (0.9)	1 (1.1)	1.000
Mask Ventilation	5 (2.4)	1 (0.9)	4 (4.3)	0.177
Cardiovascular events				
Bradycardia	3 (3.2)	0 (0)	3 (1.5)	0.090
Hypotension	16 (7.8)	7 (6.2)	9 (9.7)	0.436

RAE, respiratory adverse events; BIS, bispectral index. * denotes statistical significance with *p* value < 0.05.

**Table 3 diagnostics-13-01789-t003:** Events after the dental procedure.

Post-Procedural AE	Total Number (%) (N = 206)	BIS Group (%) (N = 113)	Non-BIS (%) (N = 93)	*p* Value
Toothache	87 (42.2)	44 (38.9)	43 (46.2)	0.323
Fever	28 (13.6)	16 (14.2)	12 (12.9)	0.841
Sleepiness	21 (10.2)	10 (8.8)	11 (11.8)	0.497
Dizziness	14 (6.8)	7 (6.1)	7 (7.5)	0.784
PONV	6 (2.9)	2 (1.8)	4 (4.3)	0.413

AE, adverse events; PONV, postoperative nausea and vomiting; BIS, bispectral index.

**Table 4 diagnostics-13-01789-t004:** Drug requirements and time to awakening/discharge.

	Total Number (N = 206)	BIS Group (N = 113)	Non-BIS (N = 93)	*p* Value
Total propofol dosage (mg)	341 ± 135	338 ± 132	334 ± 139	0.774
Propofol (mg/kg)	21.2 ± 7.3	21.2 ± 7.7	21.1 ± 6.9	0.906
Propofol (mg/kg/min)	0.178 ± 0.032	0.177 ± 0.035	0.180 ± 0.029	0.426
Time to awaken (min)	31.3 ± 17.2	30.9 ± 16.9	31.8 ± 17.6	0.715
Time to discharge (min)	68.4 ± 23.6	63.4 ± 23.2	74.5 ± 24.0	0.001 *

BIS, bispectral index. * denotes statistical significance with *p* value < 0.05.

## Data Availability

Data available upon request.

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
