# Peer review of "Application of Bispectral Index System (BIS) Monitor to Ambulatory Pediatric Dental Patients under Intravenous Deep Sedation"

_diagnostics, 2023, doi:10.3390/diagnostics13101789_

Round 1

Reviewer 1 Report

Abstract: 

Purpose -   Would rephrase that the main purpose in providing sedation is to allow dental restoration while avoiding aspiration and laryngospasm.  Respiratory depression that does not affect oxygenation or carbon dioxide retention is not deleterious.  Delayed recovery impacts clinic efficiency but is not in and of itself considered an adverse event.

Methods – Need to define how deep sedation was defined.  Was RASS score or similar used to document depth of sedation in control and intervention groups

Results: how were hypoxia and apnea defined?  Cough is not typically an adverse event but a complication that can prolong a procedure

Conclusion: Did not establish if BIS score corresponded to an accepted method of scoring level of sedation

Introduction

Would recommend eliminating the term conscious sedation and use minimal sedation instead

Line 79 – would recommend rather than catastrophic sequelae that the practitioner needs to be able to recognize and manage these known effects.

Line 82 – the authors have misquoted Dr. Cote.  Many of the deaths reported by Dr. Cote resulted from long-acting medications and insufficient monitoring.

Line 92 – Recommend adding procedural sedation scale developed by Dr. Cravero.

Line 97 – Need to add information on whether this method has been validated against the sedation scales.  As the authors point out, there is wide variability in the effective dose in children of the same age.

Methods

Would recommend dividing the groups further.  There may be significant differences between pre-school children, who cannot understand the procedure versus school age children.  No mention of whether children randomized.

Need to look at differences in adverse events based upon pre-medication received.  Chloral hydrate has a much longer half-life and could influence post-procedure recovery time.

Standard definition for apnea is a pause greater than 20 seconds not 10 seconds.

Results

Was the any correlation in these receiving ketamine and subsequent laryngospasm?

Need further details on why 2 children required endotracheal intubation.

The observed difference of post-recovery time of 10 minutes is statistically significant but not a meaningful clinical difference.

Discussion

The section on toothache could be eliminated since not pertinent to the stated study objective.

Conclusion

Would add that although outpatient dental sedation can be performed safely, the provider needs to be prepared for endotracheal intubation when necessary.

Syntax and readability was good

Reviewer 2 Report

Manuscript of considerable interest for the dental sector, need for a major revision before proceeding with the evaluation of a possible publication:

Abstract: highlighting the results obtained

Keywords: add specific ones registered on MeSH

Introduction: insert which treatments these sedated patients undergo, if they are syndromic due to problems related to carious or periodontal processes? few collaborators?

Describe the incidence of caries already studied by Prof Butera, published in CHLIDREN mdpi

Materials and methods: how was the sample size calculated?

Use the flowchart from the official consort flow chart.

Results: very confusing, reorganize the tables and better highlight the statistically significant data.

Discussion: add as future goals the use of natural substances, probiotics, paraprobiotics, postbiotics and ozonated gels to rebalance the oral microbiota and reduce the bacterial load during pre and post sedation procedures, as already studied by Prof Scribante's research group.

Conclusions: reformulate them based on the comments entered by adding a proactive action

Bibliography: add references required

Review grammar, and scientific terms

Round 2

Reviewer 2 Report

THE MANUSCRIPT HAS BEEN SUCCESSFULLY REVIEWED, IT CAN BE PUBLISHED